# Contrast-Induced Encephalopathy in Patients with Chronic Kidney Disease and End-Stage Kidney Disease: A Systematic Review and Meta-Analysis

**DOI:** 10.3390/medicines10080046

**Published:** 2023-08-08

**Authors:** Paul W. Davis, Pajaree Krisanapan, Supawit Tangpanithandee, Charat Thongprayoon, Jing Miao, Mohamed Hassanein, Prakrati Acharya, Michael A. Mao, Iasmina M. Craici, Wisit Cheungpasitporn

**Affiliations:** 1Division of Nephrology and Hypertension, Mayo Clinic, Rochester, MN 55905, USA; davis.paul3@mayo.edu (P.W.D.); pajaree_fai@hotmail.com (P.K.); supawit_d@hotmail.com (S.T.); charat.thongprayoon@gmail.com (C.T.); miao.jing@mayo.edu (J.M.); craici.iasmina@mayo.edu (I.M.C.); 2Division of Nephrology, Thammasat University Hospital, Pathum Thani 12120, Thailand; 3Division of Nephrology, University of Mississippi Medical Center, Jackson, MS 58866, USA; mhassanein@umc.edu; 4Division of Nephrology, Texas Tech Health Sciences Center El Paso, El Paso, TX 10641, USA; prakrati.c.acharya@gmail.com; 5Division of Nephrology and Hypertension, Mayo Clinic, Jacksonville, FL 32224, USA; mao.michael@mayo.edu

**Keywords:** contrast-induced encephalopathy (CIE), chronic kidney disease (CKD), end-stage kidney disease (ESKD), contrast medium neurotoxicity, dialysis and CIE recovery

## Abstract

**Background:** Contrast-induced encephalopathy (CIE) is an infrequent but serious neurological condition that occurs shortly after the administration of contrast during endovascular and angiography procedures. Patients suffering from chronic kidney disease (CKD) or end-stage kidney disease (ESKD) are considered to be at a higher risk of contrast medium neurotoxicity, due to the delayed elimination of the contrast medium. However, the occurrence and characteristics of CIE in CKD/ESKD patients have not been extensively investigated. **Methods:** We conducted a comprehensive literature search, utilizing databases such as MEDLINE, EMBASE, the Cochrane Central Register of Controlled Trials, and the Cochrane Database of Systematic Reviews, up to September 2022. The purpose was to identify documented cases of CIE among patients with CKD or ESKD. Employing a random-effects model, we calculated the pooled incidence and odds ratio (OR) of CIE in CKD/ESKD patients. **Results:** Our search yielded a total of eleven articles, comprising nine case reports and two observational studies. Among these studies, 2 CKD patients and 12 ESKD patients with CIE were identified. The majority of the CKD/ESKD patients with CIE (93%) had undergone intra-arterial contrast media and/or endovascular procedures to diagnose acute cerebrovascular disease, coronary artery disease, and peripheral artery disease. The male-to-female ratio was 64%, and the median age was 63 years (with an interquartile range of 55 to 68 years). In the two observational studies, the incidence of CIE was found to be 6.8% in CKD patients and 37.5% in ESKD patients, resulting in a pooled incidence of 16.4% (95% CI, 2.4%–60.7%) among the CKD/ESKD patients. Notably, CKD and ESKD were significantly associated with an increased risk of CIE, with ORs of 5.77 (95% CI, 1.37–24.3) and 223.5 (95% CI, 30.44–1641.01), respectively. The overall pooled OR for CIE in CKD/ESKD patients was 32.9 (95% CI, 0.89–1226.44). Although dialysis prior to contrast exposure did not prevent CIE, approximately 92% of CIE cases experienced recovery after undergoing dialysis following contrast exposure. However, the effectiveness of dialysis on CIE recovery remained uncertain, as there was no control group for comparison. **Conclusions:** In summary, our study indicates an association between CIE and CKD/ESKD. While patients with CIE showed signs of recovery after dialysis, further investigations are necessary, especially considering the lack of a control group, which made the effects of dialysis on CIE recovery uncertain.

## 1. Introduction

Contrast-induced encephalopathy (CIE) is a serious and rare complication that may occur after diagnostic imaging with contrast media [1,2]. It can cause neurological symptoms, like confusion, seizures, and an altered mental status, usually appearing within a few hours to days of contrast agent injection [3,4]. Although imaging tests such as MRI or CT can reveal cortical or subcortical edema and contrast enhancement, the primary basis for CIE diagnosis is clinical presentation [5]. The mechanism of CIE is not entirely clear, but it is believed to result from a toxic or anaphylactic reaction to the contrast agent that disrupts the blood–brain barrier, leading to neuroinflammation and brain dysfunction. The likelihood of CIE is increased by various risk factors, including pre-existing kidney disease, diabetes, hypotension, dehydration, and high doses of contrast agents [6,7,8].

The incidence of CIE is challenging to estimate due to limited large-scale studies, but the medical literature suggests that it is less than 0.1% in patients undergoing contrast-enhanced imaging procedures [2,9,10,11,12,13,14,15,16,17]. However, patients with chronic kidney disease (CKD), end-stage kidney disease (ESKD), and those undergoing dialysis are at a higher risk of developing CIE due to an impaired kidney function [2]. This impaired function can lead to contrast agent accumulation in the body, increasing the probability of adverse effects [10]. Unfortunately, the incidence of CIE in these patient populations is not well-established due to a lack of data. Although there is limited information available on mortality rates related to CIE, research has indicated that CIE is a relatively uncommon occurrence, and its symptoms are typically reversible with prompt diagnosis and proper treatment [9,11,12,13,14,15,16,17]. However, in severe cases, the risk of permanent neurological damage or even death exists. Furthermore, patients with underlying kidney disease or kidney failure are at a higher risk of adverse outcomes, including mortality, as they may be more vulnerable to the toxic effects of contrast media [3,10,11].

Given the current limited data on CIE in CKD and ESKD patients, conducting a systematic review is necessary. Thus, this systematic review was performed to assess the incidence and risk of CIE among CKD and ESKD patients.

## 2. Materials and Methods

### 2.1. Information Sources and Search Strategy

A systematic literature search was conducted by using the MEDLINE, EMBASE, Cochrane Central Register of Controlled Trials, and Cochrane Database of Systematic Reviews databases from their inception to September 2022. The search was conducted using the following terms: “contrast-induced encephalopathy” OR “CIE” OR “contrast media toxicity” AND “chronic kidney disease” OR “CKD” OR “end-stage kidney disease” OR “ESKD” OR “dialysis”. Additional studies were identified by manually searching the reference lists of the included studies and relevant reviews. The PRISMA (Preferred Reporting Items for Systematic Reviews and Meta-Analysis) [18] Statement (Appendix A) guided this study’s execution. This study provides access to the data supporting its findings via the Open Science Framework (https://osf.io/f6w7k/ (accessed on 10 June 2023)).

#### 2.1.1. Ovid MEDLINE Search

The exploration in Ovid MEDLINE was performed by employing a blend of MeSH terminology and associated keywords. The utilized search terms consisted of “contrast-induced encephalopathy” OR “CIE” OR “contrast media toxicity”, in conjunction with “chronic kidney disease” OR “CKD” OR “end-stage kidney disease” OR “ESKD” OR “dialysis”. The MeSH phrases were broadened to include all pertinent subheadings and were linked to their respective keywords. To ensure a thorough search, there were no restrictions on the language or publication date. In addition, the “related articles” feature was used to enhance the scope of the search.

#### 2.1.2. EMBASE Search

For EMBASE, the search was performed using Emtree terms corresponding to the MeSH terms used in the MEDLINE search, augmented with other relevant keywords. The Emtree terms were expanded to cover all the more specific terms. The search procedure included the following: “contrast-induced encephalopathy” OR “CIE” OR “contrast media toxicity” AND “chronic kidney disease” OR “CKD” OR “end-stage kidney disease” OR “ESKD” OR “dialysis”. Neither the language nor the date of publication were restricted.

#### 2.1.3. Cochrane Database of Systematic Reviews Search

A similar strategy was applied to search the Cochrane Database of Systematic Reviews. The search terms included the following: “contrast-induced encephalopathy” OR “CIE” OR “contrast media toxicity” AND “chronic kidney disease” OR “CKD” OR “end-stage kidney disease” OR “ESKD” OR “dialysis”. This search was not restricted by language or date to ensure the inclusiveness of all the relevant reviews.

In all the databases, the search terms were combined using appropriate Boolean operators (AND, OR). In each case, the search strategy was designed to be as comprehensive as possible, to ensure all the relevant studies were captured for further screening and potential inclusion in this review. Additionally, the reference lists of all the retrieved articles were manually scanned to identify further potentially relevant studies that were not indexed in the searched databases.

### 2.2. Selection Criteria

Studies reporting cases of CIE in CKD or ESKD patients who received contrast media during endovascular or angiography procedures were included in this study. Our study utilized a range of article types, including case reports, case series studies, observational studies, and, if available, clinical trials, as a part of the inclusion criteria. The primary objective was to gather comprehensive information on CIE in patients with CKD or ESKD who received contrast media during endovascular or angiography procedures. This diverse inclusion was imperative, due to the rarity of CIE in this specific patient population. By incorporating these diverse article types, our study aimed to gather a comprehensive range of information on CIE in CKD or ESKD patients who underwent endovascular or angiography procedures with contrast media. This comprehensive approach was indispensable for gaining a better understanding of the condition, identifying risk factors, improving diagnostic methods, and exploring potential interventions or preventive measures to mitigate the occurrence of CIE in this specific patient population. Exclusion criteria included studies that did not report the outcomes in CKD or ESKD patients, studies that did not use contrast media during endovascular or angiography procedures, studies that were not published in English, and studies with duplicate data.

### 2.3. Data Abstraction

Two independent reviewers (P.W.D. and W.C.) screened the titles and abstracts of all the studies identified by the search strategy. Potentially eligible studies were assessed for eligibility by retrieving and assessing their full text articles. The data extracted from eligible studies included the number of CKD/ESKD patients with CIE, the type of contrast media used, the type of endovascular or angiography procedures performed, and the outcomes of CIE. Discrepancies between the two reviewers were resolved by a discussion and consensus. 

### 2.4. Analysis of Bias Risk

#### 2.4.1. Bias Risk in Case Reports

During our investigation, the case reports underwent an assessment process using the JBI Critical Appraisal Checklist for Case Reports [19]. This tool provides a structured framework to judge the quality of case reports, commencing with a deep dive into the patient’s demographics. An exemplary report would delve into a detailed description of the patient, encompassing factors like age, gender, ethnicity, and other patient-specific attributes. The subsequent stage of evaluation concerns the depth of the explanation given to the condition or ailment being investigated. It is essential that the disease is expounded in a detailed manner, granting the readers a thorough understanding of the condition being reported. The next essential criterion is the exhaustive narration of the patient’s medical history. After this, the emphasis of evaluation shifts to the degree of diagnostic certainty. An exceptional case report should convincingly substantiate the precision of the diagnosis, ideally using empirical diagnostic norms [19,20,21,22,23,24,25].

The checklist then proceeds to examine the descriptions of the management and interventions employed. These descriptions should be unbiased and extensive, providing a clear understanding of the patient’s treatment strategy. Changes in the patient’s status post-intervention should also be recorded thoroughly. The following point of evaluation focuses on outcomes. These outcomes should be clearly delineated, quantifiable, and relevant to the context. Lastly, the checklist assesses the detailed follow-up information, providing insights into the patient’s status after the intervention over a suitable duration.

#### 2.4.2. Bias Risk in Case Series

For case series assessments, the NIH Quality Assessment Tool for Case Series Studies was employed [26]. The initial criterion for evaluation is the explicitness of the study objectives, which should be intrinsically linked to the research question and articulated clearly.

The next evaluation step assesses the comprehensiveness of the case series. It is vital for the case series to be all-inclusive and representative of the relevant population. Data collection should preferably be prospective rather than retrospective, ensuring the data’s accuracy and impartiality. Consistency in data collection methods across all cases is a further vital factor. The outcome measures should be reliable and valid, accurately reflecting the study findings. In addition, the legitimacy of the statistical analysis techniques is evaluated to ensure the correct interpretation of the data. The duration of follow-up is then gauged for sufficiency. The follow-up span should be adequate to identify any intervention effects. If patients are lost to follow-up, the report should provide an explanation for the same. Finally, the study should identify and adjust for potential confounding factors, leading to a more accurate and reliable conclusion.

#### 2.4.3. Bias Risk in Observational Studies

For observational studies, the validated Newcastle–Ottawa quality assessment scale [27] was utilized. This scale assesses studies across three principal domains.

The first domain, Selection, examines the representativeness of the exposed cohort, the selection of the non-exposed cohort, a verification of exposure, and affirmation that the outcome was not present at the commencement of the study. The second domain, Comparability, evaluates the cohorts’ likeness based on the study design or analysis, checking whether the researchers have taken into account other factors that could possibly influence the outcome. Lastly, the Outcome domain emphasizes the quality of the outcome measurements. It checks whether the study employed independent blind assessment or record linkage, whether the follow-up period was sufficient for the outcomes to materialize, and whether the study accounted for all the subjects in their analysis.

### 2.5. Statistical Analysis

In our meta-analysis of the observational studies, we aimed to assess the relationship between CKD or ESKD and the occurrence of CIE. The outcome of interest, CIE, was infrequent in the included studies, resulting in a low incidence or prevalence in both CKD/ESKD patients and non-CKD/non-ESKD patients. Using the risk ratio (RR) to measure the relative risk of CIE in the exposed group (CKD or ESKD patients) compared to the unexposed group (non-CKD or non-ESKD patients) becomes challenging when the event is rare. In such cases, RR estimates tend to be imprecise, leading to inflated confidence intervals, which can undermine result reliability. To address this issue, we opted to use the Odds Ratio (OR) as a valid substitute for RR when dealing with low-event rates. The OR represents the ratio of the odds of CIE occurrence in CKD/ESKD patients to the odds of CIE occurrence in non-CKD/non-ESKD patients. Since the event is rare, the OR approximates the RR and provides a reasonably accurate estimate of the relative risk, making it a suitable alternative in these circumstances [28]. The incidence and OR were presented with 95% confidence intervals (CI). The odds ratio (OR) was computed using a designated formula, contrasting the likelihood of CIE occurrence in CKD or ESKD patients versus non-CKD or non-ESKD patients. This formula took the following form: OR = ratio of CIE occurrence in CKD or ESKD patients/ratio of CIE occurrence in non-CKD or non-ESKD patients. To integrate the calculated odds ratios from each study in the meta-analysis, we applied a random-effects model due to a high I^2^ value (>50%). This model was favored, due to its ability to accommodate the heterogeneity of our study pool, as it considers variations in study characteristics, populations, and other contributing factors. It does this by acknowledging that the studies included can be a random sample with differing true effect sizes. Contrarily, a fixed-effect model, which presumes a uniform true effect size across all studies, was deemed less applicable considering the broad scope of the studies in our meta-analysis. Consequently, the pooled odds ratios allowed for a more accurate representation of the link between CKD or ESKD and the likelihood of CIE. The degree of heterogeneity among the studies was evaluated using the I^2^ statistic [29]. An eager regression symmetry test was used to assess for a publication bias [30]. A *p* value < 0.05 was considered statistically significant for all the analyses. Comprehensive Meta-Analysis software version 3.3.070 (Biostat, Englewood, NJ, USA) was utilized to carry out all the statistical analyses. 

## 3. Results

Eleven articles (nine case reports and two cohort studies) [2,9,10,11,12,13,14,15,16,17,31] with 2 CKD patients and 12 ESKD patients with CIE were identified (Figure 1 and Table 1). The studies spanned a period from 1996 to 2022, indicating the enduring relevance of CIE in patients with CKD and ESKD. 

In this study, 64% of the participants were male, and the median age was 63 years with an interquartile range (IQR) of 55 to 68 years. Most of the patients had comorbidities, such as HTN, CAD, and DM. These studies displayed an assortment of contrasting profiles, ranging from coronary angiography to cerebral angiogram and endovascular thrombectomy. We observed a diversity in the contrasts’ administration routes, which are predominantly intra-arterial (IA). The majority of patients with CKD or ESKD who experienced CIE (93%) received intra-arterial contrast media and/or underwent endovascular procedures to diagnose conditions such as acute cerebrovascular disease, coronary artery disease, and peripheral artery disease. The contrast media used were also varied, including agents like iodixanol, ioversol, and iopromide, among others. The volume of contrast administered varied substantially among the cases, ranging from 12 mL to 910 mL. Examining the encephalopathy characteristics following contrast use, we found the symptoms ranged from altered consciousness (e.g., confusion, somnolence, and loss of consciousness) to physical manifestations such as seizures, hemiparesis, blindness, and headaches.

Two cohort studies revealed that the incidence of CIE in CKD patients was 6.8%, whereas in ESKD patients, it was 37.5%. The overall pooled incidence of CIE in CKD/ESKD patients was 16.4% (95% CI, 2.4%–60.7%, I^2^ = 75%), as depicted in Figure 2A. The meta-analysis demonstrated a significantly elevated risk of CIE in both CKD and ESKD patients, with an OR of 5.77 (95% CI, 1.37–24.3) and 223.5 (95% CI, 30.44–1641.01), respectively (Figure 2B). The overall pooled OR for CIE in CKD/ESKD patients was 32.9 (95% CI, 0.89–1226.44, I^2^ = 88%). Because of the limited number of studies included in this analysis, a funnel plot was not created. Generally, tests for funnel plot asymmetry should only be used when there are at least 10 study groups. With the limited number of studies, the power of the test is too low to accurately distinguish between chance and actual asymmetry [32].

Most of the patients underwent hemodialysis (HD) soon after the onset of encephalopathy (within 0 to 5 days). The treatment response was generally positive, with most patients showing an improvement or resolution of symptoms. However, some cases resulted in severe outcomes, including multiple bilateral infarctions and one death. In the reported cases, the administration of dialysis before contrast exposure did not serve as a preventive measure for CIE. Nonetheless, among the CIE cases, approximately 92% exhibited signs of recovery when dialysis was employed following contrast exposure. However, the impact of dialysis on CIE recovery remains uncertain due to the absence of a control group in this study.

The risk of bias was evaluated in the incorporated studies on CIE in patients with CKD and ESKD. This evaluation was based on each study’s design, as indicated in Table 2. Starting with the case reports, they were assessed using the JBI Critical Appraisal Checklist for Case Reports. The reports included the studies conducted by Muruve et al. (1996) [11], Ozelsancak et al. (2010) [13], Yan et al. (2013) [15], Olbrich et al. (2017) [12], Yen et al. (2017) [16], Simsek et al. (2019) [14], Bender et al. (2020) [9], Alshaer et al. (2022) [17], and Fong et al. (2022) [31]. All of these studies were subjected to the systematic approach provided by the checklist. On the other hand, observational studies were assessed based on the validated Newcastle–Ottawa quality assessment scale. The observational studies included studies by Matsubara et al. (2017) [10] and Chu et al. (2020) [2]. These studies were evaluated across three principal domains: Selection, Comparability, and Outcome.

Each study underwent comprehensive risk bias analysis, based on the specific tool. Important aspects such as patient demographics, disease description, medical history, diagnosis certainty, management and intervention descriptions, post-intervention changes, outcomes, and follow-up information were evaluated in the case reports. In the observational studies, evaluation covered the representativeness of the cohort, outcome absence at the start of the study, similarity of cohorts, and quality of outcome measurements.

The above table provides a summary of the bias risk assessment, categorized by each study. Most studies showed a low risk of bias, indicating robust and reliable findings. However, a few case reports had a moderate risk, suggesting the need for a careful interpretation of those specific reports. Overall, the risk of bias was well handled in the majority of the studies, providing confidence in the review results.

## 4. Discussion

Our systematic reviews showed that the incidence of CIE among CKD or ESKD patients was 16.4%. The pooled odds ratio (OR) for CIE among CKD or ESKD patients was 32.9. In the reported cases, the administration of dialysis before contrast exposure did not serve as a preventive measure for CIE. The outcomes of CIE depend on the severity of symptoms and the promptness of diagnosis and treatment. For mild cases, supportive care may result in a complete recovery, whereas severe cases may lead to permanent neurological damage or even death. 

The outcomes of CIE depend on the severity of the symptoms and the promptness of diagnosis and treatment. For mild cases, supportive care such as a close monitoring of vital signs and managing seizures or neurological symptoms may result in a complete recovery [33]. On the other hand, severe cases may lead to permanent neurological damage or even death. Patients with underlying kidney disease or kidney failure may be more vulnerable to the harmful effects of contrast media and therefore, have a higher risk of adverse outcomes [2,9,10,11,12,13,14,15,16,17].

Despite its infrequent occurrence, CIE is a severe complication of contrast agent administration [2,10]. The treatment of CIE in patients with CKD and ESKD depends on the severity of the symptoms and underlying causes [2]. Typically, the initial step is to discontinue the use of contrast media and provide supportive care for symptom management. For mild cases, a close monitoring of vital signs, fluids and electrolyte balance, and management of seizures or neurological symptoms may be sufficient. In severe cases, hospitalization may be required to provide more intensive care, including intravenous fluids, electrolyte replacement, and medications for symptom management. In ESKD patients on dialysis, treatment may also include dialysis to remove excess contrast media [2,10].

The best approach to managing CIE in CKD and ESKD patients is prevention [34]. Preventing CIE involves identifying the patients at risk, minimizing the use of high doses of contrast agents, and opting for alternative imaging methods when possible, such as non-contrast MRI [34]. Close monitoring and a prompt recognition of CIE are crucial for timely management and optimal outcomes [35]. This includes a careful evaluation of the patient’s medical history and underlying risk factors before administering contrast media, the use of alternative imaging methods when possible, and the adjustment of the contrast dose or timing of the imaging procedures as necessary to minimize the risk of adverse effects [35]. Close monitoring for the signs and symptoms of CIE is also crucial in these populations to enable an early intervention and optimize the outcomes [36]. The treatment of CIE focuses on supportive care, including hydration and seizure control and the discontinuation of the contrast agent. In severe cases, corticosteroids or other immunomodulatory agents may be considered [37]. Most patients recover fully within a few days to weeks, but some may experience lasting cognitive impairment [2,9,10,11,12,13,14,15,16,17]. CIE is usually associated with iodinated contrast agents used in medical imaging, and it is not typically linked to gadolinium-based contrast agents (GBCAs) [38]. However, there have been reports of neurological symptoms occurring after the administration of GBCAs, especially in patients with pre-existing kidney disease or other risk factors. The underlying mechanism by which GBCAs may lead to neurological symptoms is not yet fully understood and is currently being studied [38,39,40].

Contrast media, which include ionic, non-ionic, low osmolarity, iso-osmolar, and high osmolarity solutions have been reported to induce CIE [41,42]. Ionic contrast agents have a higher osmolality and carry a greater risk of CIE compared to non-ionic contrast agents, which have a lower osmolality. However, both ionic and non-ionic contrast agents have been associated with CIE [41,42]. The contrast medium has a half-life of around 2 h in patients with normal kidney function [10,43,44,45,46]. However, in patients with severe kidney dysfunction, the half-life is significantly longer, exceeding 16 h [10,43,44,45,46]. In patients undergoing hemodialysis, the half-life of the contrast medium is even more prolonged. Hemodialysis can effectively eliminate contrast media from the bloodstream, with around 80% of the contrast agent being removed within four hours [10,43,44,45,46]. Peritoneal dialysis is not as efficient as hemodialysis in eliminating IV contrast media from the bloodstream. The extent of the removal of contrast media by peritoneal dialysis is influenced by various factors, including the type and dosage of the contrast media used and the patient’s peritoneal transport properties [43]. Although some studies have suggested that peritoneal dialysis can eliminate a limited amount of contrast media [43,47], it is generally not considered the preferred approach for treating CIE.

Deciphering the findings of our research necessitates an awareness of the key limitations. One primary constraint relates to the relatively small sample sizes in the studies incorporated in this review, which constricts the broad applicability of our findings. Our investigation was largely informed by case reports, which possess inherent limitations in establishing causality and providing substantial statistical significance. Moreover, there is a potential selection bias within the observational studies, as patients who underwent endovascular procedures may not adequately reflect the overall CKD or ESKD populations. Furthermore, we identified an inconsistency in the definitions and diagnostic criteria used for CIE across the studies, which contributes to a variability in the reported incidence rates. This extends to the lack of uniformity in the dosages and types of contrast media employed in different medical procedures [48,49], which can sway both the incidence and severity of CIE. These factors present a challenge in collating and synthesizing the data, complicating the process of making accurate comparisons. Future investigations could improve the credibility of these results by employing larger, more diverse sample sizes, ensuring standardized definitions and diagnostic criteria for CIE and maintaining consistent dosages and types of contrast media across diverse procedures. Longitudinal studies could yield valuable insights into the long-term impacts of CIE on patient outcomes, such as the risk of recurrent stroke or mortality. Additionally, exploring potential risk factors for CIE, like underlying comorbidities or medications, could assist in identifying high-risk patients and informing preventive strategies.

While acknowledging the constraints of our research, it is essential to underscore the numerous significant merits it encompasses. Firstly, our inclusive approach to source selection offers an expansive insight into the existing body of research on CIE. This synthesis, which embraces both small-scale case studies and more extensive observational reports, allows us to discern overarching themes and patterns not immediately apparent in standalone studies. Secondly, our meticulous dissection of the available literature has facilitated the spotlighting of the critical need for uniformity in CIE diagnosis and contrast media administration, thereby making a valuable contribution to the ongoing discourse on research methodologies within this sphere. Thirdly, our examination of the heterogeneity in study designs, sample sizes, and definition use across the range of studies provides a nuanced critique of the present research landscape. This review can act as a springboard for future research, pointing scholars towards facets warranting further exploration and enhancement. Finally, our research has ignited discussions regarding potential risk factors and the enduring impacts of CIE, laying the groundwork for potential avenues of future research. We have illuminated the need for larger, more varied sample sizes, diversified study designs, and the honing of definitions and procedures in CIE-focused studies. Therefore, our study, in spite of its limitations, offers a significant contribution to the discipline by pinpointing potential avenues for progress and establishing a resilient basis for forthcoming investigations.

## 5. Conclusions

CIE is an infrequent yet significant adverse effect stemming from endovascular procedures and angiographies. Patients with CKD or ESKD may exhibit an increased susceptibility, due to the slower processing and elimination of contrast agents. A systematic evaluation revealed that the pooled incidence of CIE in patients with CKD or ESKD stands at approximately 16.4%. This study also highlighted that individuals with CKD or ESKD are more likely to develop this condition. However, the limited number of studies, small sample sizes, and lack of standardization in the definitions and criteria used to diagnose CIE and the dosages and types of contrast media used in different medical procedures are the major limitations of this study. Future studies with larger sample sizes, standardized definitions and criteria, and uniform dosages and types of contrast media are needed to better understand the incidence and risk factors of CIE in CKD/ESKD patients and to inform preventive strategies.

## Figures and Tables

**Figure 1 medicines-10-00046-f001:**
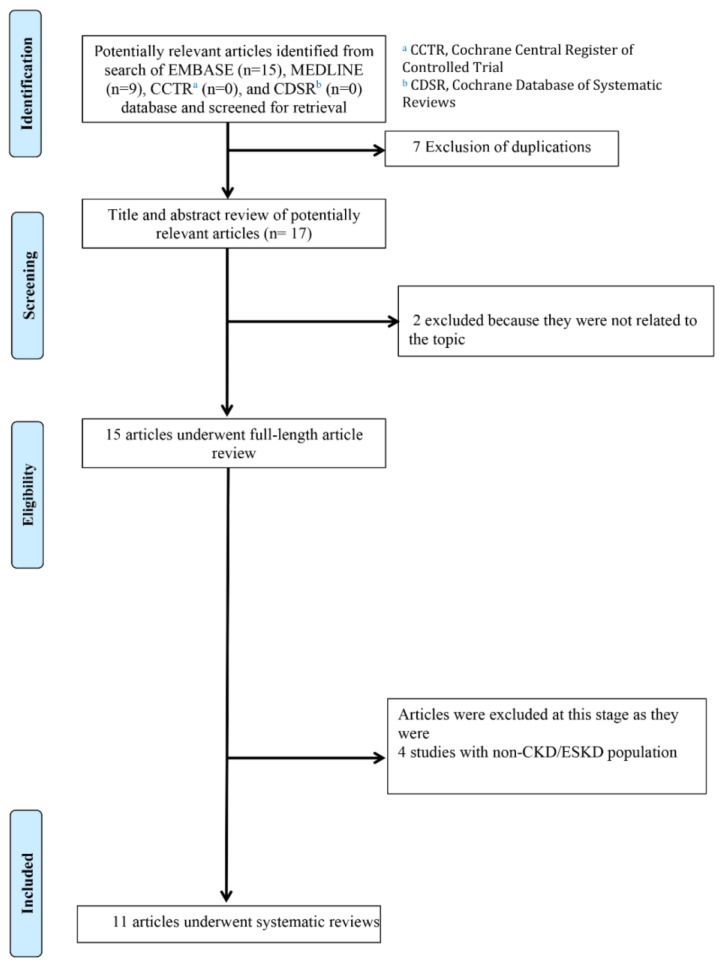
Search strategy.

**Figure 2 medicines-10-00046-f002:**
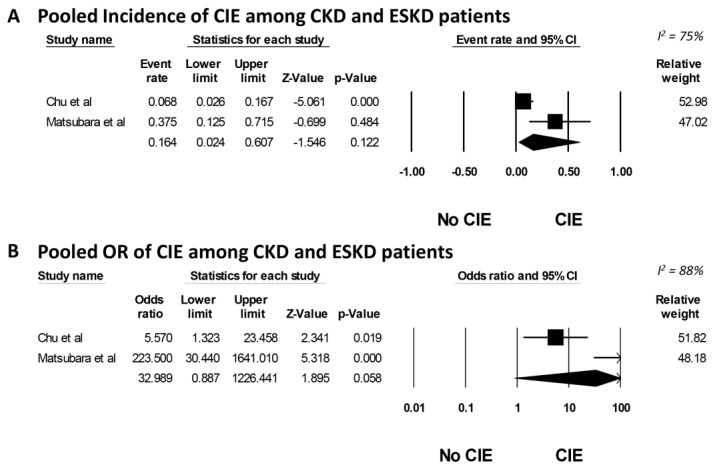
(**A**) Forest plot of meta-analysis on the incidence of CIE in CKD or ESKD patients. The diamond represents the pooled incidence with its 95% CI. CI: confidence interval; CIE: contrast-induced encephalopathy; CKD: chronic kidney disease; ESKD: end-stage kidney disease; and M–H: Mantel–Haenszel. (**B**) Forest plot of meta-analysis on OR of CIE in CKD or ESKD patients. The plot shows the OR (log scale) against its standard error (SE). CIE: contrast-induced encephalopathy; CKD: chronic kidney disease; ESKD: end-stage kidney disease; OR: odds ratio; and SE: standard error [2,10].

**Table 1 medicines-10-00046-t001:** Characteristics of included studies of contrast-induced encephalopathy in patients with CKD and ESKD.

Author (Year)	Muruve et al. (1996) [11]	Ozelsancak et al. (2010) [13]	Yan et al. (2013) [15]	Olbrich et al. (2017) [12]	Yen et al.(2017) [16]	Simsek et al. (2019) [14]	Bender et al. (2020) [9]	Alshaer et al. (2022) [17]	Fong et al. (2022) [31]	Matsubara et al. (2017) [10]	Matsubara et al. (2017) [10]	Matsubara et al. (2017) [10]	Chu et al. (2020) [2]	Chu et al. (2020) [2]
Publication type	Case report	Case report	Case report	Case report	Case report	Case report	Case report	Case report	Case report	Cohort study	Cohort study	Cohort study	Cohort study	Cohort study
Age (year)	49	55	63	76	64	68	46	80 s	28	63	74	63	84	61
Comorbidities	Unstable angina, HTN, chronic glomerular nephritis	DM, HTN, CAD	HTN, DM, rheumatoid arthritis	CAD	HTN, CAD, DM	DM, HTN, CAD	HTN, hypothyroidism, PKD		Lupus nephritis	HTN, PKD	HTN, liver cirrhosis, mitral insufficiency, DM	HTN, PKD	HTN, recent myocardial infarct	HTN, HLD, HF, AF, prior ipsilateral MCA stroke
Baseline kidney function (CKD, ESRD)	ESKD	ESKD	ESKD	CKD	ESKD	CKD4	ESKD	ESKD	ESKD	ESKD	ESKD	ESKD	ESKD	ESKD
Contrast profile	Coronary angiography	Abdominal angiography	Cerebral angiogram	Coronary angiography	Peripheral CT angiogram	Coronary angiography	Cerebral angiography	CTA head	CT angiogram and CT brain	Left internal carotid angiogram with coil embolization	Endovascular procedure for basilar tip, unruptured	Endovascular procedure for basilar tip, unruptured	Endovascular thrombectomy	Endovascular thrombectomy
Name	Meglumine/sodium diatrizoate	Ioversol	Iodixanol (Visipaque 320)	-	Iobitridol	Iohexol (Biemexol 300)	Iopamiro 370	-	-	Iodixanol (Visipaque 270)	Ipamidol (Iopamilon 300)	Ipamidol (Iopamilon 300)	Iopromide	Iopromide
Route (IV, IA)	IA	IA	IA	IA	IA	IA	IA	IA	IV/IA	IA	IA	IA	IA	IA
Dose (mL)	90, 610	100	910	280	100, 150	230	80	-	-	210	160	300	42	12
Encephalopathy characteristic	Headache, seizures	Confusion, agitation	Somnolence and reduced spontaneous movement	LOC	Irritation, disorientation, anisocoria	Seizures	Blurring of vision to blindness, seizure	Seizures	LOC	Hemiparesis, convulsion	Blindness, consciousness disturbance	Blindness	Worsened NIHSS	Worsened NIHSS
Dialysis mode	HD	HD	HD	HD	HD	HD	HD	-	HD	HD	HD	HD	HD	HD
Days dialysis after encephalopathy	1	0	0	5	0	1	2	-	-	1	1	0	-	-
Response to treatment	Improvement	Improvement	Improvement after weaning sedation on day 3	Multiple bilateral infarctions	Improvement	Improvement	Complete resolution	-	Improvement	Improvement	Resolution	Improvement	Deceased	-
Onset of improvement (day)	1	15	3	-	0	1	5	1	After > 1 HD session	1	1	-	-	-

Abbreviations: AKI: acute kidney injury, CAD: coronary artery disease, CIE: contrast-induced encephalopathy, CKD: chronic kidney disease, CT: computed tomography, DM: diabetes mellitus, EEG: electroencephalogram, ESKD: end-stage kidney disease, ESRD: end-stage renal disease, GFR: glomerular filtration rate, HD: hemodialysis, HTN: hypertension, IA: intra-arterial, ICA: internal carotid artery, LOC: loss of consciousness, MR: magnetic resonance, N/A: not available, PCI: percutaneous coronary intervention, PD: peritoneal dialysis.

**Table 2 medicines-10-00046-t002:** Summary of bias risk assessment.

Author (Year)	Study Design	Risk of Bias
Muruve et al. (1996) [11]	Case report	Low
Ozelsancak et al. (2010) [13]	Case report	Moderate
Yan et al. (2013) [15]	Case report	Low
Olbrich et al. (2017) [12]	Case report	Low
Yen et al. (2017) [16]	Case report	Moderate
Simsek et al. (2019) [14]	Case report	Low
Bender et al. (2020) [9]	Case report	Low
Alshaer et al. (2022) [17]	Case report	Moderate
Fong et al. (2022) [31]	Case report	Moderate
Matsubara et al. (2017) [10]	Cohort study	Low
Chu et al. (2020) [2]	Cohort study	Low

Note: The ‘Risk of Bias’ assessment in this table is representative. Actual bias levels should be deduced from the thorough bias risk analysis based on the JBI checklist and Newcastle–Ottawa scale.

## Data Availability

Upon reasonable request, the authors are willing to share the data.

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
