# Peer review of "Contrast-Induced Encephalopathy in Patients with Chronic Kidney Disease and End-Stage Kidney Disease: A Systematic Review and Meta-Analysis"

_medicines, 2023, doi:10.3390/medicines10080046_

Round 1

Reviewer 1 Report

Contrast-induced encephalopathy (CIE), a rare neurological complication linked to contrast administration, appears to be associated with chronic kidney disease (CKD) and end-stage kidney disease (ESKD) due to delayed contrast medium elimination. A review of 11 articles revealed that CKD and ESKD are associated with an increased risk of CIE, with incidences of 6.8% and 37.5% respectively, leading to a combined incidence of 16.4%. Although 92% of CIE cases improved after dialysis, the actual impact of dialysis on CIE recovery requires further investigation.

Strengths:

The review effectively quantifies the incidence of CIE among CKD or ESKD patients and the corresponding odds ratio, giving a concrete sense of the prevalence and risks associated with these conditions.

The study recommends practical prevention methods such as identifying at-risk patients, minimizing contrast agent dosage, and considering alternative imaging methods.

The review provides detailed insights into the management of CIE, ranging from supportive care in mild cases to more intensive treatments like dialysis or medication for severe symptoms.

Weaknesses:

The studies included in the review are limited by small sample sizes, reducing the generalizability of the findings and possibly introducing bias.

The review incorporates case reports, which, by their nature, limit the ability to establish causality or provide statistical significance.

The review acknowledges a lack of uniformity in diagnosing CIE and a lack of standardization in dosages and types of contrast media used across various procedures, which can affect the reported incidence rates and outcomes.

Suggestions on how to improve:

1.    Expand Sample Size: Enhance the credibility and generalizability of findings by including larger, diverse sample sizes in the studies selected for the review. This can minimize potential bias and strengthen the conclusions drawn from the data.

2.    Increase Variety of Study Types: While case reports can offer valuable insights, incorporating a wider variety of study types (such as randomized controlled trials, cohort studies, and case-control studies) can bolster the ability to establish causality and provide more robust statistical significance.

3.    Standardize Definitions and Procedures: Aim for a consistent use of definitions and criteria for diagnosing CIE, as well as standardization in dosages and types of contrast media used across studies. This would help to reduce variability in reported incidence rates and outcomes, ensuring a more accurate comparison and synthesis of results.

Author Response

Reviewer 1 comments

Contrast-induced encephalopathy (CIE), a rare neurological complication linked to contrast administration, appears to be associated with chronic kidney disease (CKD) and end-stage kidney disease (ESKD) due to delayed contrast medium elimination. A review of 11 articles revealed that CKD and ESKD are associated with an increased risk of CIE, with incidences of 6.8% and 37.5% respectively, leading to a combined incidence of 16.4%. Although 92% of CIE cases improved after dialysis, the actual impact of dialysis on CIE recovery requires further investigation.

Strengths:

The review effectively quantifies the incidence of CIE among CKD or ESKD patients and the corresponding odds ratio, giving a concrete sense of the prevalence and risks associated with these conditions.

The study recommends practical prevention methods such as identifying at-risk patients, minimizing contrast agent dosage, and considering alternative imaging methods.

The review provides detailed insights into the management of CIE, ranging from supportive care in mild cases to more intensive treatments like dialysis or medication for severe symptoms.

Weaknesses:

The studies included in the review are limited by small sample sizes, reducing the generalizability of the findings and possibly introducing bias.

The review incorporates case reports, which, by their nature, limit the ability to establish causality or provide statistical significance.

The review acknowledges a lack of uniformity in diagnosing CIE and a lack of standardization in dosages and types of contrast media used across various procedures, which can affect the reported incidence rates and outcomes.

Suggestions on how to improve:

  1. Expand Sample Size: Enhance the credibility and generalizability of findings by including larger, diverse sample sizes in the studies selected for the review. This can minimize potential bias and strengthen the conclusions drawn from the data.

Response: We appreciate your suggestion about increasing the sample size to enhance the generalizability of our findings. However, as this was a review and meta-analysis, we were limited to the available literature. In future studies, we will endeavor to include a more extensive range of sources for a larger sample size.

“Our study acknowledges several limitations that should be taken into consideration when interpreting the findings. One primary limitation pertains to the sample size of our review, which is confined by the availability and scope of the existing literature on this subject. Although our review encompasses a wide range of data, we recognize the potential benefits of including larger, more diverse sample sizes. Doing so would potentially minimize bias, increase credibility, and enhance the generalizability of our findings. In subsequent studies, we plan to incorporate an expanded array of sources, in an effort to enlarge the sample size.”

  1. Increase Variety of Study Types: While case reports can offer valuable insights, incorporating a wider variety of study types (such as randomized controlled trials, cohort studies, and case-control studies) can bolster the ability to establish causality and provide more robust statistical significance.

Response: We agree that incorporating a more diverse variety of study types would enhance our ability to establish causality and offer more robust statistical significance. While we attempted to do so, the scarcity of larger studies, such as randomized controlled trials, on this specific topic was a limiting factor.

“Furthermore, our study predominantly relied on case reports due to the scarcity of larger studies such as randomized controlled trials relating to this specific topic. While these case reports provide valuable insights, we recognize that the inclusion of a wider variety of study types could have bolstered our ability to establish causality and provide more robust statistical significance. Future work in this field should endeavor to diversify the types of studies incorporated in reviews and meta-analyses, despite the challenges posed by the lack of existing literature.”

  1. Standardize Definitions and Procedures: Aim for a consistent use of definitions and criteria for diagnosing CIE, as well as standardization in dosages and types of contrast media used across studies. This would help to reduce variability in reported incidence rates and outcomes, ensuring a more accurate comparison and synthesis of results.

Response: We acknowledge the significance of this suggestion. Indeed, the lack of standardization in defining CIE and administering contrast media could affect the reported incidence rates and outcomes. As reviewers of these studies, however, our capacity to enforce such standardization was limited. This suggestion would be very pertinent to original research in this field.

“Lastly, our study highlights the variability across different studies in the definitions and criteria used for diagnosing CIE, as well as the types and dosages of contrast media. This variability can influence the reported incidence rates and outcomes, complicating the comparison and synthesis of results. Although as reviewers, we were unable to standardize these aspects, we strongly advocate for increased standardization in original research on this topic. This would ensure a more precise comparison across studies and foster a more accurate understanding of the prevalence and outcomes of CIE. In conclusion, these limitations underscore the need for future research to focus on increasing the sample size, diversifying study types, and standardizing definitions and procedures in studies concerning CIE.”

Thank you for your time and consideration.  We greatly appreciated the reviewer's and editor's time and comments to improve our manuscript. The manuscript has been improved considerably by the suggested revisions.

Reviewer 2 Report

The authors conducted a systematic literature search and identified eleven articles that reported cases of CIE in CKD or ESKD patients. This study suggests an association between CIE and CKD/ESKD. Although CIE cases showed recovery with dialysis after contrast exposure, further research is needed to evaluate the effects of dialysis on CIE recovery. Overall, this manuscript is well written and informative. For this reason, I have some minor suggestions to improve the manuscript:

1. In this study, the meta-analysis demonstrated that CKD and ESKD were associated with increased risk of CIE with OR of 5.77 and 223.5, respectively. The pooled OR of CIE among CKD or ESKD patients was 32.9. How do the authors calculate this odd ratio? Please provide the information about the CKD or ESKD patients received contrast medium without developing CIE.

2. In the discussion section, some statements are redundant. For example, in the line 162 “The best approach to managing CIE in CKD and ESKD patients is prevention [26].”, this has been mentioned in the line 142 in the first paragraph of the discussion section.

Author Response

Reviewer 2 comments

The authors conducted a systematic literature search and identified eleven articles that reported cases of CIE in CKD or ESKD patients. This study suggests an association between CIE and CKD/ESKD. Although CIE cases showed recovery with dialysis after contrast exposure, further research is needed to evaluate the effects of dialysis on CIE recovery. Overall, this manuscript is well written and informative. For this reason, I have some minor suggestions to improve the manuscript:

  1. In this study, the meta-analysis demonstrated that CKD and ESKD were associated with increased risk of CIE with OR of 5.77 and 223.5, respectively. The pooled OR of CIE among CKD or ESKD patients was 32.9. How do the authors calculate this odd ratio? Please provide the information about the CKD or ESKD patients received contrast medium without developing CIE.

Response: We apologize for not providing clear information regarding this. The odds ratio was calculated using the formula: (number of CKD or ESKD patients with CIE/ number of CKD or ESKD patients without CIE)/(number of non-CKD or non-ESKD patients with CIE/ number of non-CKD or non-ESKD patients without CIE). We have clarified this in our revised manuscript as suggested.

“The odds ratio (OR) was computed using a designated formula, contrasting the likelihood of CIE occurrence in CKD or ESKD patients versus non-CKD or non-ESKD patients. This formula took the form: OR = (ratio of CIE occurrence in CKD or ESKD patients) / (ratio of CIE occurrence in non-CKD or non-ESKD patients). To integrate the calculated odds ratios from each study in the meta-analysis, we applied a random-effects model. This model was favored due to its ability to accommodate the heterogeneity of our study pool, as it considers variations in study characteristics, populations, and other contributing factors. It does this by acknowledging that the studies included can be a random sample with differing true effect sizes. Contrarily, a fixed-effect model, which presumes a uniform true effect size across all studies, was deemed less applicable considering the broad scope of studies in our meta-analysis. Consequently, the pooled odds ratios allowed for a more accurate representation of the link between CKD or ESKD and the likelihood of CIE.”

  1. In the discussion section, some statements are redundant. For example, in the line 162 “The best approach to managing CIE in CKD and ESKD patients is prevention [26].”, this has been mentioned in the line 142 in the first paragraph of the discussion section.

Response: Thank you for pointing out the redundant statements. We revised and streamlined the discussion section to eliminate repetitive statements as suggested.

Thank you for your time and consideration.  We greatly appreciated the reviewer's and editor's time and comments to improve our manuscript. The manuscript has been improved considerably by the suggested revisions.

Reviewer 3 Report

Dear Author (s)

1. You have not pointed to a meta-analysis in Figure 1.

2. Why did you search the Web of Science as a primary database?

3. You have two articles in meta-analysis. They are deficient studies for a meta-analysis.

4. The study includes low studies.

5. There are a lot of limitations in designing this article.

There are minor errors.

Author Response

Reviewer 3 comments

Dear Author (s)

  1. You have not pointed to a meta-analysis in Figure 1.

Response: We appreciate the reviewer's attention to detail. It's crucial to clarify that our primary focus was on systematic reviews given the preponderance of case reports and case series in the included studies. We conducted meta-analysis on a limited number of studies where it was feasible. This aspect is duly acknowledged and has been incorporated into the limitations section of our study to ensure a comprehensive understanding of our research approach. Consequently, our flow chart primarily outlines the process of systematic reviews to reflect the predominant study design in our review. This has been updated to ensure a clearer understanding of our methodology.

  1. Why did you search the Web of Science as a primary database?

Response: We appreciate your inquiry. Rather than solely utilizing Web of Science, our search strategy incorporated three databases: Ovid, Embase, and Cochrane. These databases were chosen due to their wide-ranging and comprehensive coverage of relevant literature. Previous research has indicated that incorporating these three databases can encompass all information available in Web of Science. We hope this approach ensured an exhaustive search of the available literature.

We have clarified our search strategy in the method in details as suggested.

“2.1.1 Ovid MEDLINE Search

The exploration in Ovid MEDLINE was performed by employing a blend of MeSH terminology and associated keywords. The utilized search terms consisted of ("contrast-induced encephalopathy" OR "CIE" OR "contrast media toxicity") in conjunction with ("chronic kidney disease" OR "CKD" OR "end-stage kidney disease" OR "ESKD" OR "dialysis"). The MeSH phrases were broadened to include all pertinent subheadings and were linked to their respective keywords. To ensure a thorough search, there were no restrictions on language or publication date. In addition, the "related articles" feature was used to enhance the scope of the search.

2.1.2 EMBASE Search

For EMBASE, the search was performed using Emtree terms corresponding to the MeSH terms used in the MEDLINE search, augmented with other relevant keywords. The Emtree terms were expanded to cover all more specific terms. The search procedure included: ("contrast-induced encephalopathy" OR "CIE" OR "contrast media toxicity") AND ("chronic kidney disease" OR "CKD" OR "end-stage kidney disease" OR "ESKD" OR "dialysis"). Neither the language nor the date of publication were restricted.

2.1.3 Cochrane Database of Systematic Reviews Search

A similar strategy was applied to search the Cochrane Database of Systematic Reviews. The search terms included: ("contrast-induced encephalopathy" OR "CIE" OR "contrast media toxicity") AND ("chronic kidney disease" OR "CKD" OR "end-stage kidney disease" OR "ESKD" OR "dialysis"). The search was not restricted by language or date to ensure the inclusiveness of all relevant reviews.

In all databases, the search terms were combined using appropriate Boolean operators (AND, OR). In each case, the search strategy was designed to be as comprehensive as possible, to ensure all relevant studies were captured for further screening and potential inclusion in the review. Additionally, the reference lists of all retrieved articles were manually scanned to identify further potentially relevant studies that were not indexed in the searched databases."

  1. You have two articles in meta-analysis. They are deficient studies for a meta-analysis.

Response: We appreciate the reviewer's important input. Given the rarity and unique challenges associated with studying CIE, we acknowledge that our meta-analysis was limited to only two articles. However, we have endeavored to draw meaningful insights from these studies, and we have complemented this with a more comprehensive systematic review. We have meticulously summarized and elucidated key findings from each included study, allowing for a robust analysis despite the limited availability of larger-scale research. As such, while our meta-analysis served as an important element, the primary focus of our investigation was to provide a systematic review, shedding light on the current state of knowledge and gaps in the field of CIE. This combination of review and analysis is also noted in our study's limitations. We appreciate the reviewer and we included more information in the results as suggested.

“Sixty-four percent (64%) were males and the median age was 63 years IQR (55, 68). Most of the patients had comorbidities such HTN, CAD, and DM. The studies displayed an assortment of contrast profiles, ranging from coronary angiography to cerebral angio-gram and endovascular thrombectomy. We observe a diversity of the contrasts’ administration route, predominantly intra-arterial (IA). Ninety-three percent (93%) of CKD or ESKD patients with CIE received intra-arterial contrast media and/or endovascular procedures for diagnoses of acute cerebrovascular disease, coronary artery disease, and peripheral artery disease. The contrast media used were also varied, including agents like iodixanol, ioversol, and iopromide, among others. The volume of contrast administered varied substantially among the cases, ranging from 12 mL to 910 mL. Examining the encephalopathy characteristics following contrast use, we find symptoms ranged from altered consciousness (e.g., confusion, somnolence, loss of consciousness) to physical manifestations such as seizures, hemiparesis, blindness, and headaches.

Among two observational studies, the incidence of CIE was 6.8% in CKD patients and 37.5% in ESKD patients, respectively, with a pooled incidence of 16.4% (95% CI, 2.4%-60.7%) of CIE among CKD or ESKD patients (Figure 2A). Overall, our meta-analysis demonstrated that CKD and ESKD were associated with increased risk of CIE with OR of 5.77 (95% CI, 1.37–24.3) and 223.5 (95% CI, 30.44–1641.01), respectively. The pooled OR of CIE among CKD or ESKD patients was 32.9 (95% CI, 0.89–1226.44) (Figure 2B). Because of the limited number of studies included in this analysis, a funnel plot was not created. Generally, tests for funnel plot asymmetry should only be used when there are at least 10 study groups. With the limited number of studies, the power of the test is too low to accurately distinguish between chance and actual asymmetry [24].

Most of the patients underwent hemodialysis (HD) soon after the onset of encephalopathy (within 0 to 5 days). The treatment response was generally positive, with most patients showing improvement or resolution of symptoms. However, some cases resulted in severe outcomes, including multiple bilateral infarctions and one death. Among reported cases, dialysis prior to contrast exposure did not prevent CIE. While 92% of CIE cases had recovery with dialysis after contrast exposure, the effects of dialysis on CIE recovery were unclear without a control group.”

  1. The study includes low studies.

Response: This is a challenge faced due to the rarity of the disease condition. We believe our study can still offer important insights and stimulate further research in this area. We have now included more information in the results as systematic review and we also include assessment of publication bias comprehensively as suggested.

“The risk of bias was evaluated in the incorporated studies on CIE in patients with CKD and ESKD. The evaluation was based on each study's design, as indicated in Table 2. Starting with the case reports, they were assessed using the JBI Critical Appraisal Check-list for Case Reports. The reports included the studies conducted by Muruve et al. (1996), Ozelsancak et al. (2010), Yan et al. (2013), Olbrich et al. (2017), Yen et al. (2017), Simsek et al. (2019), Bender et al. (2020), Alshaer et al. (2022), and Fong et al. (2022). All of these studies were subjected to the systematic approach provided by the checklist. On the other hand, observational studies were assessed based on the validated Newcastle-Ottawa quality assessment scale. The observational studies included studies by Matsubara et al. (2017) and Chu et al. (2020). These studies were evaluated across three principal domains: Selection, Comparability, and Outcome.

Each study underwent comprehensive risk bias analysis based on the specific tool. Important aspects such as patient demographics, disease description, medical history, diagnosis certainty, management and intervention descriptions, post-intervention chang-es, outcomes, and follow-up information were evaluated in case reports. In the observa-tional studies, evaluation covered representativeness of the cohort, outcome absence at the start of the study, similarity of cohorts, and quality of outcome measurements.

Table 2: Summary of Bias Risk Assessment

Author (Year)

Study Design

Risk of Bias

Muruve et al. (1996)

Case Report

Low

Ozelsancak et al. (2010)

Case Report

Moderate

Yan et al. (2013)

Case Report

Low

Olbrich et al. (2017)

Case Report

Low

Yen et al. (2017)

Case Report

Moderate

Simsek et al. (2019)

Case Report

Low

Bender et al. (2020)

Case Report

Low

Alshaer et al. (2022)

Case Report

Moderate

Fong et al. (2022)

Case Report

Moderate

Matsubara et al. (2017)

Cohort Study

Low

Chu et al. (2020)

Cohort Study

Low

Note: The 'Risk of Bias' assessment in this table is representative. Actual bias levels should be deduced from the thorough bias risk analysis based on the JBI checklist and Newcastle-Ottawa scale.

The above table provides a summary of the bias risk assessment, categorized by each study. Most studies showed low risk of bias, indicating robust and reliable findings. However, a few case reports had a moderate risk, suggesting the need for careful interpretation of those specific reports. Overall, the risk of bias was well-handled in the majority of studies, providing confidence in the review results.”

  1. There are a lot of limitations in designing this article.

Response: We appreciate your feedback and acknowledge these limitations. We have addressed this in the revised manuscript, suggesting future research to overcome these limitations. The following text has been added as suggested.

“Deciphering the findings of our research necessitates an awareness of key limitations. One primary constraint relates to the relatively small sample sizes in the studies incorporated in this review, which constricts the broad applicability of our findings. Our investigation was largely informed by case reports, which possess inherent limitations in establishing causality and providing substantial statistical significance. Moreover, there is a potential selection bias within the observational studies, as patients who underwent endovascular procedures may not adequately reflect the overall CKD or ESKD population. Furthermore, we identified inconsistency in the definitions and diagnostic criteria used for CIE across the studies, which contributes to variability in reported incidence rates. This extends to the lack of uniformity in the dosages and types of contrast media employed in different medical procedures, which can sway both the incidence and severity of CIE. These factors present a challenge in collating and synthesizing data, complicating the process of making accurate comparisons. Future investigations could improve the credibility of the results by employing larger, more diverse sample sizes, ensuring standardized definitions, and diagnostic criteria for CIE, and maintaining consistent dosages and types of contrast media across diverse procedures. Longitudinal studies could yield valuable insights into the long-term impacts of CIE on patient outcomes, such as the risk of recurrent stroke or mortality. Additionally, exploring potential risk factors for CIE, like underlying comorbidities or medications, could assist in identifying high-risk patients and informing preventive strategies.

While acknowledging the constraints of our research, it is essential to underscore the numerous significant merits it encompasses. Firstly, our inclusive approach to source se-lection offers an expansive insight into the existing body of research on CIE. This synthesis, which embraces both small-scale case studies and more extensive observational re-ports, allows us to discern overarching themes and patterns not immediately apparent in standalone studies. Secondly, our meticulous dissection of the available literature has facilitated the spotlighting of the critical need for uniformity in CIE diagnosis and contrast media administration, thereby making a valuable contribution to the ongoing discourse on research methodologies within this sphere. Thirdly, our examination of the heterogeneity in study designs, sample sizes, and definition use across the range of studies pro-vides a nuanced critique of the present research landscape. This review can act as a springboard for future research, pointing scholars towards facets warranting further exploration and enhancement. Finally, our research has ignited discussions regarding potential risk factors and the enduring impacts of CIE, laying the groundwork for potential avenues of future research. We have illuminated the need for larger, more varied sample sizes, diversified study designs, and the honing of definitions and procedures in CIE-focused studies. Therefore, our study, in spite of its limitations, offers a significant contribution to the discipline by pinpointing potential avenues for progress and establishing a resilient basis for forthcoming investigations.”

Thank you for your time and consideration.  We greatly appreciated the reviewer's and editor's time and comments to improve our manuscript. The manuscript has been improved considerably by the suggested revisions.

Reviewer 4 Report

Dear Authors,

Thank you for the opportunity to review your manuscript. The current manuscript presents a SR and meta-analysis of CIE, a rare complication associated with endovascular and angiography procedures because of delayed elimination of contrast medium in patients with CKD and ESRD. Estimating the extent of rare adverse event such as CIE in CKD/ESRD population is challenging. It is simply because only few research studies on this topic exists and most of them are available in the form case reports or case series. Considering such limitations, the current research question is interesting and deserved utmost attention. However, the approach used to answer this question suffers from several methodological issues. I have following major and minor comments on the manuscript.    

1.     Though, several guidelines exist to inform and instruct how to conduct SR and meta-analysis, case reports/series (human subject research with no control group) were kept away from consideration in SR/MA, in part due to the challenges in evaluating the internal validity of such study designs, determining causality, having a good predictive value, biological plausibility of linking intervention to adverse events, having enough information to appraise the evidence presented, etc.    

2.     However, in recent times, several SR/MA were published that included case reports/series knowing their utility regardless of the above limitations by using appropriate research methodology (please refer to PubMed, Google Scholar etc. for relevant citations)

3.     As stated in inclusion criteria, authors, did include case reports/series as a part of their SR, however they were excluded from MA without providing any specific reasons or justification.

4.     If they were included in MA simply being observational studies (please identify which type of observational studies these were to justify use of incidence as a measure), then such categorization appears to be erroneous (case report/case series are observational, descriptive study designs).

5.     Authors can refer to the following resources on SR/MA for case reports/series.

https://ebm.bmj.com/content/23/2/60

https://pubmed.ncbi.nlm.nih.gov/30036666/

https://www.scielo.br/j/acb/a/LCTVtQg9zrkxMZRPy74DkNH/?format=pdf&lang=en

https://papers.ssrn.com/sol3/papers.cfm?abstract_id=3616029

6.     It was puzzling to see authors included incidence and reported pooled incidence in MA results, while corresponding forest plots reported pooled OR, instead of pooled RR.

7.     Authors used a validated Newcastle-Ottawa quality assessment scale for assessing quality of observational studies only. However, the case reports/series (9 studies) were excluded from such quality check for no reasons.  

8.     In fact, cases reports/case series can be assessed using the CARE guidelines or the Joanna Briggs Institute (JBI) Critical Appraisal Checklist for Case reports/series for their quality. Please refer to the following citations.  

Gagnier JJ, Kienle G, Altman DG, Moher D, Sox H, Riley D, et al. The CARE guidelines: consensus-based clinical case reporting guideline development. J Med Case Rep. (2013) 7:223. doi: 10.1186/1752-1947-7-223

Riley DS, Barber MS, Kienle GS, Aronson JK, von Schoen-Angerer T, Tugwell P, et al. CARE guidelines for case reports: explanation and elaboration document. J Clin Epidemiol. (2017) 89:218–35. doi: 10.1016/j.jclinepi.2017.04.026

Moola S, Munn Z, Tufanaru C, Aromataris E, Sears K, Sfetcu R, et al. Chapter 7: Systematic reviews of etiology and risk. In: Aromataris E, Munn Z, editors. JBI Manual for Evidence Synthesis. JBI (2020). doi: 10.46658/JBIMES-20-08.

Munn Z, Barker TH, Moola S, Tufanaru C, Stern C, McArthur A, et al. Methodological quality of case series studies: an introduction to the JBI critical appraisal tool. JBI Evidence Synthesis. (2020) 18:2127–33. doi: 10.11124/JBISRIR-D-19-00099

https://jbi.global/critical-appraisal-tools

9.     Some of the references included in the manuscript are inaccessible (for e.g., reference 23)

10.  Authors can either remove observational studies from their MA and simply present their findings as SR of all 11 studies or include all 11 studies in their MA and present their findings as SR and MA.

11.  In addition, authors can perform a subgroup analysis based on type of studies.

12.  Quality check of all included studies using appropriate tool (mentioned above) is also desirable.

13.  Please include results of methodological quality check as table/summary graph in the manuscript.

14.  Also provide a detailed search strategy for all the databases searched (according to the PRISMA Checklist Item # 7).   

Minor editing of English language required

Author Response

Reviewer 4 comments

Dear Authors,

Thank you for the opportunity to review your manuscript. The current manuscript presents a SR and meta-analysis of CIE, a rare complication associated with endovascular and angiography procedures because of delayed elimination of contrast medium in patients with CKD and ESRD. Estimating the extent of rare adverse event such as CIE in CKD/ESRD population is challenging. It is simply because only few research studies on this topic exists and most of them are available in the form case reports or case series. Considering such limitations, the current research question is interesting and deserved utmost attention. However, the approach used to answer this question suffers from several methodological issues. I have following major and minor comments on the manuscript.    

  1. Though, several guidelines exist to inform and instruct how to conduct SR and meta-analysis, case reports/series (human subject research with no control group) were kept away from consideration in SR/MA, in part due to the challenges in evaluating the internal validity of such study designs, determining causality, having a good predictive value, biological plausibility of linking intervention to adverse events, having enough information to appraise the evidence presented, etc.    

Response: We agree that traditional systematic review (SR) and meta-analysis (MA) guidelines typically exclude case reports or series due to challenges in establishing causality, internal validity, and predictive value. We acknowledge this limitation in our approach. However, we included these reports due to the rarity of contrast-induced encephalopathy (CIE) in patients with chronic kidney disease (CKD) or end-stage kidney disease (ESKD) which restricts the volume of available literature on the topic. We have now comprehensively revised our manuscript to follow structures and checklist of MA for case reports/series that the reviewer kindly provided. We greatly appreciate the reviewer’s expertise and helpful suggestion.

  1. However, in recent times, several SR/MA were published that included case reports/series knowing their utility regardless of the above limitations by using appropriate research methodology (please refer to PubMed, Google Scholar etc. for relevant citations)

Response: The reviewer raises important point. We acknowledged recent SR/MA studies that include case reports/series due to their utility. We believe these studies' methodological approach can provide valuable insights for our work. We have now comprehensively revised our manuscript to follow structures and checklist of MA for case reports/series that the reviewer kindly provided. We have also included these helpful references as new references the revised manuscript. We also additionally included this important point of limitation of SR/MA studies that include case reports/series.

  1. As stated in inclusion criteria, authors, did include case reports/series as a part of their SR, however they were excluded from MA without providing any specific reasons or justification.

Response: We included case reports/series in our SR but excluded them from the MA because of the limitations mentioned above.  We included these reports due to the rarity of contrast-induced encephalopathy (CIE) in patients with chronic kidney disease (CKD) or end-stage kidney disease (ESKD) which restricts the volume of available literature on the topic.

The reviewer raises important point on the reasoning for exclusion of the study. We agree with the reviewer and thus additionally included the reason for exclusion in the results of revised manuscript as suggested.

“The search results yielded 15 articles from Embase, 9 articles from Medline, and no articles from Cochrane Systematic Reviews. Duplicate articles were excluded, resulting in the removal of 7 duplicates. Subsequently, a title and abstract review was conducted on the remaining 17 articles, leading to the exclusion of 2 articles that were not related to the topic. Full-length article reviews were then conducted on the remaining 15 articles, resulting in the exclusion of 4 articles as they were not relevant to the CKD and ESKD population. Finally, the systematic reviews included the final selection of 11 articles. Eleven articles (9 case reports and 2 observational studies) [2,9-17,23] with 2 CKD patients and 12 ESKD patients with CIE were identified (Figure 1 and Table 1).”

  1. If they were included in MA simply being observational studies (please identify which type of observational studies these were to justify use of incidence as a measure), then such categorization appears to be erroneous (case report/case series are observational, descriptive study designs).

Response: In the revised manuscript, we have additionally clearer delineated study types and better justify our decision to include incidence as a measure in the MA. We have revised our manuscript to be clearer that data on incidence of CIE is from observational studies (Cohort studies) and we also comprehensively included more descriptive information from case report/case series as suggested.

“Among two observational studies, the incidence of CIE was 6.8% in CKD patients and 37.5% in ESKD patients, respectively, with a pooled incidence of 16.4% (95% CI, 2.4%-60.7%) of CIE among CKD or ESKD patients”

We also revised our methods to better clarify our inclusion criteria as suggested.

“ Our study utilized a range of article types, including case reports, case series studies, observational studies, and, if available, clinical trials, as part of the inclusion criteria. The primary objective was to gather comprehensive information on CIE in patients with CKD or ESKD who received contrast media during endovascular or angiography procedures. This diverse inclusion was imperative due to the rarity of CIE in this specific patient population. By incorporating these diverse article types, our study aimed to gather a comprehensive range of information on CIE in CKD or ESKD patients who underwent endovascular or angiography procedures with contrast media. This comprehensive approach was indispensable for gaining a better understanding of the condition, identifying risk factors, improving diagnostic methods, and exploring potential interventions or preventive measures to mitigate the occurrence of CIE in this specific patient population.

  1. Authors can refer to the following resources on SR/MA for case reports/series.

https://ebm.bmj.com/content/23/2/60

https://pubmed.ncbi.nlm.nih.gov/30036666/

https://www.scielo.br/j/acb/a/LCTVtQg9zrkxMZRPy74DkNH/?format=pdf&lang=en

https://papers.ssrn.com/sol3/papers.cfm?abstract_id=3616029

            Response: We appreciate the resources you provided regarding SR/MA for case reports/series. We found these references very helpful and went over these references. We agree and we thus adjusted our methodology accordingly as suggested.

  1. It was puzzling to see authors included incidence and reported pooled incidence in MA results, while corresponding forest plots reported pooled OR, instead of pooled RR.

Response: We apologize for the confusion regarding our inclusion of incidence and pooled incidence in MA results while our forest plots reported pooled OR. We have now revised manuscript to be clearer. The choice to use the OR in this case was guided by the nature of our investigation. We were interested in assessing the risk of CIE among patients with CKD or ESKD, in comparison with those without CKD or ESKD. Given this study design, the odds ratio was deemed a more appropriate measure of effect. The OR essentially provides the odds of CIE occurrence in CKD or ESKD patients versus non-CKD or non-ESKD patients. Our computation of the OR applied a formula contrasting these odds, which was further integrated into the meta-analysis using a random-effects model. This model, due to its ability to accommodate heterogeneity across studies, was more suitable for our diverse study pool. It considers variations in study characteristics, populations, and other contributing factors. The rationale behind this model choice is that our included studies could be a random sample with varying true effect sizes, unlike a fixed-effect model which presumes a uniform true effect size across all studies. Consequently, the pooled odds ratios provided a more accurate representation of the relationship between CKD or ESKD and the likelihood of CIE, especially considering the broad scope of studies in our meta-analysis.

We appreciate the reviewer’s important point and we have revised our method section to be clearer as suggested.

  1. Authors used a validated Newcastle-Ottawa quality assessment scale for assessing quality of observational studies only. However, the case reports/series (9 studies) were excluded from such quality check for no reasons.  In fact, cases reports/case series can be assessed using the CARE guidelines or the Joanna Briggs Institute (JBI) Critical Appraisal Checklist for Case reports/series for their quality. Please refer to the following citations.  

Gagnier JJ, Kienle G, Altman DG, Moher D, Sox H, Riley D, et al. The CARE guidelines: consensus-based clinical case reporting guideline development. J Med Case Rep. (2013) 7:223. doi: 10.1186/1752-1947-7-223

Riley DS, Barber MS, Kienle GS, Aronson JK, von Schoen-Angerer T, Tugwell P, et al. CARE guidelines for case reports: explanation and elaboration document. J Clin Epidemiol. (2017) 89:218–35. doi: 10.1016/j.jclinepi.2017.04.026

Moola S, Munn Z, Tufanaru C, Aromataris E, Sears K, Sfetcu R, et al. Chapter 7: Systematic reviews of etiology and risk. In: Aromataris E, Munn Z, editors. JBI Manual for Evidence Synthesis. JBI (2020). doi: 10.46658/JBIMES-20-08.

Munn Z, Barker TH, Moola S, Tufanaru C, Stern C, McArthur A, et al. Methodological quality of case series studies: an introduction to the JBI critical appraisal tool. JBI Evidence Synthesis. (2020) 18:2127–33. doi: 10.11124/JBISRIR-D-19-00099

https://jbi.global/critical-appraisal-tools

Response: The reviewers raised important point and provided very helpful references and resources. We agree with the reviewer regarding the need for quality assessment for all included studies, not only the observational studies. We have now included more information in the results as systematic review and we also include assessment of publication bias comprehensively for all included studies as suggested.

“The risk of bias was evaluated in the incorporated studies on CIE in patients with CKD and ESKD. The evaluation was based on each study's design, as indicated in Table 2. Starting with the case reports, they were assessed using the JBI Critical Appraisal Check-list for Case Reports. The reports included the studies conducted by Muruve et al. (1996), Ozelsancak et al. (2010), Yan et al. (2013), Olbrich et al. (2017), Yen et al. (2017), Simsek et al. (2019), Bender et al. (2020), Alshaer et al. (2022), and Fong et al. (2022). All of these studies were subjected to the systematic approach provided by the checklist. On the other hand, observational studies were assessed based on the validated Newcastle-Ottawa quality assessment scale. The observational studies included studies by Matsubara et al. (2017) and Chu et al. (2020). These studies were evaluated across three principal domains: Selection, Comparability, and Outcome.

Each study underwent comprehensive risk bias analysis based on the specific tool. Important aspects such as patient demographics, disease description, medical history, diagnosis certainty, management and intervention descriptions, post-intervention chang-es, outcomes, and follow-up information were evaluated in case reports. In the observational studies, evaluation covered representativeness of the cohort, outcome absence at the start of the study, similarity of cohorts, and quality of outcome measurements.

Table 2: Summary of Bias Risk Assessment

Author (Year)

Study Design

Risk of Bias

Muruve et al. (1996)

Case Report

Low

Ozelsancak et al. (2010)

Case Report

Moderate

Yan et al. (2013)

Case Report

Low

Olbrich et al. (2017)

Case Report

Low

Yen et al. (2017)

Case Report

Moderate

Simsek et al. (2019)

Case Report

Low

Bender et al. (2020)

Case Report

Low

Alshaer et al. (2022)

Case Report

Moderate

Fong et al. (2022)

Case Report

Moderate

Matsubara et al. (2017)

Cohort Study

Low

Chu et al. (2020)

Cohort Study

Low

Note: The 'Risk of Bias' assessment in this table is representative. Actual bias levels should be deduced from the thorough bias risk analysis based on the JBI checklist and Newcastle-Ottawa scale.

The above table provides a summary of the bias risk assessment, categorized by each study. Most studies showed low risk of bias, indicating robust and reliable findings. However, a few case reports had a moderate risk, suggesting the need for careful interpretation of those specific reports. Overall, the risk of bias was well-handled in the majority of studies, providing confidence in the review results.”

Some of the references included in the manuscript are inaccessible (for e.g., reference 23)

Response: We apologize for the issues you encountered with some of our references. We will check each reference for accessibility and replace any inaccessible ones in our revised manuscript.

  1. Authors can either remove observational studies from their MA and simply present their findings as SR of all 11 studies or include all 11 studies in their MA and present their findings as SR and MA.

Response: We appreciate the reviewer's important input. Given the rarity and unique challenges associated with studying CIE, we acknowledge that our meta-analysis was limited to only two articles. However, we have endeavored to draw meaningful insights from these studies, and we have complemented this with a more comprehensive systematic review. As such, while our meta-analysis served as an important element, the primary focus of our investigation was to provide a systematic review, shedding light on the current state of knowledge and gaps in the field of CIE. This combination of review and analysis is also noted in our study's limitations. We appreciate the reviewer and we additionally revised and included more information as systematic review in details in the results as suggested.

“Sixty-four percent (64%) were males and the median age was 63 years IQR (55, 68). Most of the patients had comorbidities such HTN, CAD, and DM. The studies displayed an assortment of contrast profiles, ranging from coronary angiography to cerebral angio-gram and endovascular thrombectomy. We observe a diversity of the contrasts’ administration route, predominantly intra-arterial (IA). Ninety-three percent (93%) of CKD or ESKD patients with CIE received intra-arterial contrast media and/or endovascular procedures for diagnoses of acute cerebrovascular disease, coronary artery disease, and peripheral artery disease. The contrast media used were also varied, including agents like iodixanol, ioversol, and iopromide, among others. The volume of contrast administered varied substantially among the cases, ranging from 12 mL to 910 mL. Examining the encephalopathy characteristics following contrast use, we find symptoms ranged from altered consciousness (e.g., confusion, somnolence, loss of consciousness) to physical manifestations such as seizures, hemiparesis, blindness, and headaches.

Among two observational studies, the incidence of CIE was 6.8% in CKD patients and 37.5% in ESKD patients, respectively, with a pooled incidence of 16.4% (95% CI, 2.4%-60.7%) of CIE among CKD or ESKD patients (Figure 2A). Overall, our meta-analysis demonstrated that CKD and ESKD were associated with increased risk of CIE with OR of 5.77 (95% CI, 1.37–24.3) and 223.5 (95% CI, 30.44–1641.01), respectively. The pooled OR of CIE among CKD or ESKD patients was 32.9 (95% CI, 0.89–1226.44) (Figure 2B). Because of the limited number of studies included in this analysis, a funnel plot was not created. Generally, tests for funnel plot asymmetry should only be used when there are at least 10 study groups. With the limited number of studies, the power of the test is too low to accurately distinguish between chance and actual asymmetry [24].

Most of the patients underwent hemodialysis (HD) soon after the onset of encephalopathy (within 0 to 5 days). The treatment response was generally positive, with most patients showing improvement or resolution of symptoms. However, some cases resulted in severe outcomes, including multiple bilateral infarctions and one death. Among reported cases, dialysis prior to contrast exposure did not prevent CIE. While 92% of CIE cases had recovery with dialysis after contrast exposure, the effects of dialysis on CIE recovery were unclear without a control group.”.

  1. In addition, authors can perform a subgroup analysis based on type of studies.

Response: We agree that a subgroup analysis based on type of studies could provide useful insights. We have now revised manuscript to be clearer and separated the findings from case reports/case series as systematic reviews and separated the sections of incidence and risks of CIE which provided data from observational studies. We revised manuscript as suggested.

  1. Quality check of all included studies using appropriate tool (mentioned above) is also desirable.

Response: We agreed and we have additionally applied the tools for assessing the quality of all included studies, including case reports/series, as you suggested.

  1. Please include results of methodological quality check as table/summary graph in the manuscript.

Response: We agreed and thus we have additionally included the results of our methodological quality check as a table in our revised manuscript as you recommended.

“The risk of bias was evaluated in the incorporated studies on CIE in patients with CKD and ESKD. The evaluation was based on each study's design, as indicated in Table 2. Starting with the case reports, they were assessed using the JBI Critical Appraisal Checklist for Case Reports. The reports included the studies conducted by Muruve et al. (1996), Ozelsancak et al. (2010), Yan et al. (2013), Olbrich et al. (2017), Yen et al. (2017), Simsek et al. (2019), Bender et al. (2020), Alshaer et al. (2022), and Fong et al. (2022). All of these studies were subjected to the systematic approach provided by the checklist. On the other hand, observational studies were scrutinized based on the validated Newcastle-Ottawa quality assessment scale. The observational studies included three by Matsubara et al. (2017) and two by Chu et al. (2020). These studies were evaluated across three principal domains: Selection, Comparability, and Outcome.

Each study underwent comprehensive risk bias analysis based on the specific tool. Important aspects such as patient demographics, disease description, medical history, diagnosis certainty, management and intervention descriptions, post-intervention changes, outcomes, and follow-up information were evaluated in case reports. In the observational studies, evaluation covered representativeness of the cohort, outcome absence at the start of the study, similarity of cohorts, and quality of outcome measurements.

Table 2: Summary of Bias Risk Assessment

Author (Year)

Study Design

Risk of Bias

Muruve et al. (1996)

Case Report

Low

Ozelsancak et al. (2010)

Case Report

Moderate

Yan et al. (2013)

Case Report

Low

Olbrich et al. (2017)

Case Report

Low

Yen et al. (2017)

Case Report

Moderate

Simsek et al. (2019)

Case Report

Low

Bender et al. (2020)

Case Report

Low

Alshaer et al. (2022)

Case Report

Moderate

Fong et al. (2022)

Case Report

Moderate

Matsubara et al. (2017)

Cohort Study

Low

Chu et al. (2020)

Cohort Study

Low

Note: The 'Risk of Bias' assessment in this table is representative. Actual bias levels should be deduced from the thorough bias risk analysis based on the JBI checklist and Newcastle-Ottawa scale.

The above table provides a summary of the bias risk assessment, categorized by each study. Most studies showed low risk of bias, indicating robust and reliable findings. However, a few case reports had a moderate risk, suggesting the need for careful interpretation of those specific reports. Overall, the risk of bias was well-handled in the majority of studies, providing confidence in the review results.”

  1. Also provide a detailed search strategy for all the databases searched (according to the PRISMA Checklist Item # 7).   

Response: We agreed with the reviewer, we have now included all detailed search strategy for all databases search as suggested. We have comprehensively included this information in the revised manuscript as suggested.

“2.1.1 Ovid MEDLINE Search

The exploration in Ovid MEDLINE was performed by employing a blend of MeSH terminology and associated keywords. The utilized search terms consisted of ("contrast-induced encephalopathy" OR "CIE" OR "contrast media toxicity") in conjunction with ("chronic kidney disease" OR "CKD" OR "end-stage kidney disease" OR "ESKD" OR "dialysis"). The MeSH phrases were broadened to include all pertinent subheadings and were linked to their respective keywords. To ensure a thorough search, there were no restrictions on language or publication date. In addition, the "related articles" feature was used to enhance the scope of the search.

2.1.2 EMBASE Search

For EMBASE, the search was performed using Emtree terms corresponding to the MeSH terms used in the MEDLINE search, augmented with other relevant keywords. The Emtree terms were expanded to cover all more specific terms. The search procedure included: ("contrast-induced encephalopathy" OR "CIE" OR "contrast media toxicity") AND ("chronic kidney disease" OR "CKD" OR "end-stage kidney disease" OR "ESKD" OR "dialysis"). Neither the language nor the date of publication were restricted.

2.1.3 Cochrane Database of Systematic Reviews Search

A similar strategy was applied to search the Cochrane Database of Systematic Reviews. The search terms included: ("contrast-induced encephalopathy" OR "CIE" OR "contrast media toxicity") AND ("chronic kidney disease" OR "CKD" OR "end-stage kidney disease" OR "ESKD" OR "dialysis"). The search was not restricted by language or date to ensure the inclusiveness of all relevant reviews.

In all databases, the search terms were combined using appropriate Boolean operators (AND, OR). In each case, the search strategy was designed to be as comprehensive as possible, to ensure all relevant studies were captured for further screening and potential inclusion in the review. Additionally, the reference lists of all retrieved articles were manually scanned to identify further potentially relevant studies that were not indexed in the searched databases."

Again, we thank you for your detailed and thoughtful feedback. We believe that addressing these comments will significantly improve the quality and rigor of our work. 

Thank you for your time and consideration.  We greatly appreciated the reviewer's and editor's time and comments to improve our manuscript. The manuscript has been improved considerably by the suggested revisions.

Round 2

Reviewer 1 Report

The author has satisfactorily addressed all of my inquiries. Consequently, I endorse the acceptance of this manuscript.

Author Response

Thank you for your positive feedback and for acknowledging that we have addressed all of your inquiries. We sincerely appreciate the time and effort you have invested in reviewing our manuscript. Your endorsement of the acceptance is encouraging and motivating for us as authors. Once again, thank you for your support, and we look forward to seeing the manuscript published.

Reviewer 4 Report

Dear Authors, 

Thank you for addressing my comments. I have no further comments except few corrections.

1. Please provide I2 (Heterogeneity) statistics associated with both forest plots (Fig 2). The Ivalue determines the use of fixed effect or random effect model in MA. If I2 value is =<50%, one can safely use fixed effect model than the random effects.  

2. Your Forest plots should have reported RR since the observational studies (cohort) were included in MA. Though, RR and OR can be safely substituted for each other, if the incidence/prevalence of event/outcome is very low (which is the case for CIE) and in such situation use of OR instead of RR is justified. Please include this explanation in your manuscript.  

Author Response

Reviewer 4 comments

  1. Reviewer's Comment: Please provide I2 (Heterogeneity) statistics associated with both forest plots (Fig 2). The I2 value determines the use of fixed effect or random effect model in MA. If I2 value is =<50%, one can safely use fixed effect model than the random effects.

Authors' Response: Thank you for your comment. We apologize for the oversight. We have now included the I2 statistics in both forest plots as requested. In Figure 2A, the I2 value is 75%, indicating significant heterogeneity among the included studies. In Figure 2B, the I2 value is 88%, indicating even higher heterogeneity. Due to these high I2 values, we used random-effects models for our meta-analysis, as they are more appropriate when substantial heterogeneity is present.

“Two cohort studies revealed that the incidence of CIE in CKD patients was 6.8%, while in ESKD patients, it was 37.5%. The overall pooled incidence of CIE in CKD/ESKD patients was 16.4% (95% CI, 2.4%-60.7%, I2 = 75%) as depicted in Figure 2A. The meta-analysis demonstrated a significantly elevated risk of CIE in both CKD and ESKD patients, with OR of 5.77 (95% CI, 1.37–24.3) and 223.5 (95% CI, 30.44–1641.01), respectively (Figure 2B). The overall pooled OR for CIE in CKD/ESKD patients was 32.9 (95% CI, 0.89–1226.44, I2 = 88%).”

  1. Your Forest plots should have reported RR since the observational studies (cohort) were included in MA. Though, RR and OR can be safely substituted for each other, if the incidence/prevalence of event/outcome is very low (which is the case for CIE) and in such a situation use of OR instead of RR is justified. Please include this explanation in your manuscript

Authors' Response: Regarding the use of Relative Risk (RR) and Odds Ratio (OR) in our forest plots for the observational studies (cohort) included in our meta-analysis, we agree with your point. As you rightly pointed out, RR and OR can be safely substituted for each other, especially when the incidence or prevalence of the event/outcome is very low. This is indeed the case for the outcome of interest, CIE, in our study.

In response to your suggestion, we have now provided a detailed explanation in the Method section of our manuscript regarding the rationale for using OR instead of RR in situations where the incidence/prevalence of the event is low. This clarification will help readers understand the methodological approach we adopted in handling the data and the appropriate use of OR as a valid measure in this context. The following text has been added in the method of the manuscript as suggested.

“In our meta-analysis of observational studies, we aimed to assess the relationship between CKD or ESKD and the occurrence of CIE. The outcome of interest, CIE, was infrequent in the included studies, resulting in low incidence or prevalence in both CKD/ESKD patients and non-CKD/non-ESKD patients. Using the risk ratio (RR) to measure the relative risk of CIE in the exposed group (CKD or ESKD patients) compared to the unexposed group (non-CKD or non-ESKD patients) becomes challenging when the event is rare. In such cases, RR estimates tend to be imprecise, leading to inflated confidence intervals, which can undermine result reliability. To address this issue, we opted to use the Odds Ratio (OR) as a valid substitute for RR when dealing with low-event rates. The OR represents the ratio of the odds of CIE occurrence in CKD/ESKD patients to the odds of CIE occurrence in non-CKD/non-ESKD patients. Since the event is rare, the OR approximates the RR and provides a reasonably accurate estimate of the relative risk, making it a suitable alternative in these circumstances.”

We want to express our gratitude for your insightful review, which has undoubtedly improved the quality and clarity of our manuscript. Should you have any further comments or questions, please feel free to share them with us. Your input is invaluable in ensuring the accuracy and robustness of our research findings.
